# Influence of Ag Photodeposition Conditions over SERS Intensity of Ag/ZnO Microspheres for Nanomolar Detection of Methylene Blue

**DOI:** 10.3390/nano11123414

**Published:** 2021-12-16

**Authors:** Luis Zamora-Peredo, Josué Ismael García-Ramirez, Amado Carlos García-Velasco, Julián Hernández-Torres, Leandro García-González, Monserrat Bizarro, Adriana Báez-Rodríguez

**Affiliations:** 1Centro de Investigación en Micro y Nanotecnología, Universidad Veracruzana, Adolfo Ruiz Cortines 455, Boca del Río 94294, Mexico; jonas9316@hotmail.com (J.I.G.-R.); amadocarlosgv@outlook.com (A.C.G.-V.); julihernandez@uv.mx (J.H.-T.); leagarcia@uv.mx (L.G.-G.); adbaez@uv.mx (A.B.-R.); 2Instituto de Investigaciones en Materiales, Universidad Nacional Autónoma de México, Coyoacán 04510, Mexico; monserrat@iim.unam.mx

**Keywords:** SERS, Ag/ZnO microspheres, photodeposition, rhodamine 6G, methylene blue

## Abstract

Surface enhanced Raman spectroscopy (SERS) is considered a versatile and multifunctional technique with the ability to detect molecules of different species at very low molar concentration. In this work, hierarchical ZnO microspheres (ZnO MSs) and Ag/ZnO MSs were fabricated and decorated by hydrothermal and photodeposition methods, respectively. For Ag deposition, precursor molar concentration (1.9 and 9.8 mM) and UV irradiation time (5, 15, and 30 min) were evaluated by SEM, TEM, X-ray diffraction and Raman spectroscopy. X-ray diffraction showed a peak at 37.9° corresponding to the (111) plane of Ag, whose intensity increases as precursor concentration and UV irradiation time increases. SEM images confirmed the formation of ZnO MSs (from 2.5 to 4.5 µm) building by radially aligned two-dimensional ZnO nanosheets with thicknesses below 30 nm. The Raman spectra of Ag/ZnO MSs exhibited a vibration mode at 486 cm^−1^ which can be directly associated to Ag deposition on ZnO MSs surface. The performance of SERS substrate was evaluated using rhodamine 6G. The SERS substrate grown at 9.8 mM during 30 min showed the best SERS activity and the ability to detect methylene blue at 10^−9^ M.

## 1. Introduction

As a novel sensing technique, surface-enhanced Raman scattering (SERS) has received much attention in recent years due to its ultra-sensitiveness (concentration ~10^−14^ M) to detect organic molecules such as rhodamine 6G (R6G), picric acid, crystal violet, among others [1,2,3,4,5,6]. Many SERS substrates have been proposed to enhance the Raman signals with metal nanoparticles (NPs) fixed over surface of cellulose fibers [1], metals [2], polymers [3], or metal-oxide nanostructures [4,5,6], with an extensive variety of shapes and sizes to take advantage of the higher surface-volume ratio. They are usually made in a two-step process, harmonizing chemical and physical methods to successfully synthesize a high-surface template first, and next decorate it with metal NPs. Some combinations observed are hydrothermal in the first step and then use chemical reduction in the second step [4], such as hydrothermal and in situ solution crystal growth [5], hydrothermal/magnetron sputtering [6], and solvothermal/Sn (II) ion activation [7]. However, these methodologies imply high costs, long synthesis times, high temperatures, complicated multi-steps, and the use of seed-layers, surfactants, and toxic reducing agents that are unfriendly to the environment. Considering the above, it is indispensable to develop ultra-sensitive and reusable SERS substrates obtained from simple fabrication methods, including the low costs.

Hierarchical zinc oxide microspheres (ZnO MSs) have a sphere-shaped morphology built by radially aligned two-dimensional nanosheets. These structures have proved to have powerful properties in many recent studies. Chunsheng Lei et al. reported that hierarchical porous zinc oxide microspheres (ZnO MSs) synthesized via a facile hydrothermal method have highly efficient adsorption of another organic molecule (Congo red) [8]. Jingjing Liu et al. reported a triethylamine gas sensor based on nanosized-Pt-decorated hierarchical ZnO microspheres that exhibits better sensitivity and selectivity than ZnO microspheres without Pt [9]. These hierarchical ZnO MSs offer the possibility of development template-free SERS substrates to take advantages of the highly efficient adsorption of the organic molecules by the porous microspheres and the plasmonic effects originate from the metal nanoparticles. For example, Yanjun Liu et al. detected R6G, phenol red, dopamine and glucose molecules at concentrations as low as 10^−12^ M, 10^−11^ M, 10^−12^ M, and 10^−11^ M, respectively, by SERS measurements on template-free Ag NPs-decorated ZnO microspheres (Ag/ZnO MSs) [10]. Ag/ZnO MSs-based SERS substrates are gaining more interest due to its highly sensitive capabilities, associated with recyclable and self-cleaning, suggesting potential applications as a multifunctional platform to rapid in situ monitoring photocatalytic degradation of organic pollutants in water [5,10].

Recently, Sun et al. studied silver-coated flower-like ZnO nanorod (Ag/ZnO NRs) arrays grown on texturized Si wafers enabled to detect R6G as low as 10^−14^ M and they concluded that crystallinity of silver in Ag/ZnO NRs substrates is another crucial factor of preventing from oxidizing, thus maintaining the optical properties of silver in the long-term [6]. Therefore, more detailed studies are necessary regarding Ag deposition over surface ZnO nanostructures in order to identify signal characteristics of the crystallinity of Ag NPs.

In this contribution, we report a facile and low-cost method to obtain ZnO MSs and Ag/ZnO MSs, employing hydrothermal and photodeposition methods, respectively. The hydrothermal process was chosen due to its several advantages [11], such as using low molar concentration from the precursor without using any seed layer, toxic agents, or additional procedures. For Ag NPs, the photodeposition method was selected due to its simplicity and low cost. The unique reducing agent is the UV light, i.e., Ag+ ions (AgNO_3_ as precursor) were one-step reduced-deposited on the ZnO surface when the UV irradiation was applied in just a few minutes. Morphological, structural, and optical properties of hierarchical ZnO MSs and Ag/ZnO MSs were studied by scanning electron microscopy (SEM), energy dispersive spectroscopy (EDS), transmission electron microscopy (TEM), X-ray diffraction (XRD), and Raman spectroscopy. Finally, the performance of the SERS substrate was probed with R6G and methylene blue (MB) detection.

## 2. Experimental

### 2.1. Materials

Zinc acetate dihydrate (Zn(CH_3_COO)_2_·2H_2_O); Sigma-Aldrich, St. Louis, MO, USA), sodium hydroxide (NaOH; Meyer, Mexico City, Mexico), silver nitrate (AgNO_3_; Sigma-Aldrich), deionized water and reagent grade ethanol, which were used without further purification and R6G (Sigma Aldrich).

### 2.2. Ag/ZnO Substrate Preparation

Figure 1 describes our methodology, divided into two steps. The first step involves the synthesis of the hierarchical ZnO MSs, where Zn (CH_3_COO)_2_·2H_2_O and NaOH were used as precursors, which were mixed and taken to an autoclave at 100 °C for 5.5 h, then finally cooled down to room temperature and the product was then dried. From the previous reactions, it is known that the Zn(OH)_2_ precipitate under hydrothermal process conditions will dissolve into Zn^2+^ and OH^−^ ions, so that when these ions reach the degree of supersaturation, the nucleation process occurs and ZnO crystals form. The chemical reaction mechanism carried out in this process is represented by the following reactions:$$Zn{\left(CH_{3}COO\right)}_{2}\cdot 2H_{2}O+2NaOH\;\to \;Zn{\left(OH\right)}_{2}+2CH_{3}COONa+2H_{2}O,$$
$$Zn\;{\left(OH\right)}_{2}\;\to Zn^{2+}+2OH^{-}$$
$$Zn^{2+}+2OH^{-}\;\to \;ZnO+H_{2}O$$

In the second step, for Ag deposition, a solution with 1.9 mM and 9.8 mM of AgNO_3_ diluted in deionized water and ethanol was used, adding 0.05 g of ZnO MSs, which was subsequently irradiated with a 375 nm and 3 mW laser during 5, 15, and 30 min. The Ag reduction on the ZnO MSs is originated when the electrons in valence band are excited by the laser irradiation and pass to the conduction band. Later, Ag+ ions are reduced, thus becoming a metal Ag. The Ag amount is proportional to the irradiation time and AgNO_3_ concentration.
$$ZnO+hv\;\to \;ZnO\;\left(e^{-}+h^{+}\right)$$
$$e^{-}+Ag^{+}\;\to \;Ag$$

### 2.3. Characterization

The morphology, composition, and elemental distribution of the ZnO MSs and Ag/ZnO MSs were characterized using a field emission scanning electron microscope (FE-SEM, JEOL, 7600F, Tokyo, Japan) and a transmission electron microscope (TEM, JEOL, JEM-ARM 200F, Tokyo, Japan). The crystallinity of the ZnO MSs and Ag/ZnO were made by X-ray diffraction (XRD, Bruker, D8 Advance, Karlsruhe, Germany). XRD patterns were analyzed with Panalytical X Pert HighScore Plus program. Raman spectroscopy characterization was made using a confocal Raman microscopy (Raman, Thermo Scientific, DRX, Madison, WI, USA), equipped with a 532 nm-wavelength and 10 mW-power emission laser. All Raman spectra were collected in backscattering geometry with collection of 10 points of each sample.

### 2.4. Preparation of SERS Substrate

Our SERS substrates were prepared to evaluate their SERS activity as described below. First, ZnO MSs and Ag/ZnO MSs powders were added in ethanol (mass-volume concentration: 0.05 g/mL). Then, we prepared R6G at 10^−4^ M and added 20 μL to the mixed solution. Finally, the samples were allowed to stand for 50 min without exposure to light at room temperature. The samples (the same quantity from all cases) were put on glass base to SERS measurements. All SERS spectra were collected in backscattering geometry with a collection time of 3 s and a spectral resolution of 4 cm^−1^. Three samples for each growth conditions were measured in 10 different points.

MB was selected as probe molecule to obtain the limit detection from our samples. First, the organic molecule was prepared in different molar concentrations (from 10^−6^ to 10^−9^ M). Later, the SERS powder was dissolved in MB solution (for each molar concentration) and stirred during 15 min. After precipitating, excess dye was removed and the samples were dried to room temperature for SERS measurements.

## 3. Results and Discussion

### 3.1. Scanning and Transmission Electron Microscopy (SEM and TEM)

Figure 2 shows the typical morphology of hierarchical ZnO MS obtained by the hydrothermal method. Hierarchical ZnO MS have a sphere-shaped morphology with radially aligned two-dimensional nanosheets. Microspheres possess an average diameter of 3.1 μm determined by statistical analysis (Figure 2b). The nanosheets that form the hierarchical ZnO MS can be observed in Figure 2c, in which the average thickness is 24 nm.

SEM images of ZnO MS with 1.9 and 9.8 mM of precursor concentration during different UV irradiation times (5, 15, and 30 min) can be seen in Figure 3. Once again, microspheres were obtained. Nevertheless, dispersed nanosheets can be appreciated, originating from the photodeposition process, as this behavior was observed only after Ag deposition. EDS measurement into SEM determined the Ag element presence in all samples, even though Ag nanoparticles were not possible to observe in the SEM images.

Figure 4 presents TEM images (a) and EDS mapping (b) of Ag/ZnO MSs obtained with the photodeposition process during 30 min and 9.8 mM of AgNO_3_. ZnO nanosheets have high concentration of nanopores and Ag NPs have spherical morphology with diameters ~35 nm. EDS mapping into TEM provided oxygen, zinc, and silver atoms distribution, which was useful to clearly see Ag NPs and ZnO nanosheets Ag atoms, so ZnO MS are decorated by these Ag NPs in general. The samples studied in this work have discrete Ag NPs concentration (less than 5% wt.), which allows clearly observed Ag NP by TEM and spectroscopies techniques, as follows.

### 3.2. X-ray Diffraction

The crystallinity of the hierarchical ZnO MSs and Ag/ZnO MSs was evaluated by the X-ray diffraction technique to determine the crystal size, crystalline structure, growth-direction preferential of ZnO MSs, and to compare the Ag NPs with different photodeposition-times and precursor concentrations. Figure 5 shows the XRD pattern for the pure ZnO MSs, and well-defined peaks were obtained at: 31.55°, 34.25°, 36.05°, 47.35°, 56.4°, 62.65°, 66.2°, 67.75°, 68.85°, 72.35°, and 76.75°, which correspond to the crystalline planes in (100), (002), (101), (102), (110), (103), (200), (112), (201), (004), and (202) of ZnO in the Wurtzite phase, respectively, and are well matched with the available literature (JCPDS, File No. 036-1451) [5]. From these data, the grain size was calculated using the Scherrer equation achieving a grain size of 25.6 nm. The XRD patterns of Ag/ZnO NSs have a characteristic peak at 37.9° corresponding to the (111) index of a face-centered cubic structure of Ag as per available literature (JCPDS, File No. 4-0783) [12]. The grain size for this case is 13.8 nm on average. The intensity variation of the (111) plane associated to the Ag NPs as a function of growth conditions (precursor concentration and UV irradiation time) is presented in the inset of Figure 5, where one can observe a significative change in plane intensity for 15 and 30 min of UV irradiation time in both precursor concentrations. At lower UV irradiation times, the (111) plane intensity associated to Ag structure is very low, so their values intensity are insignificant. However, as the irradiation time increases, the plane intensity increases and the full-width at half maximum decreases. This behavior is due to more crystallinity in Ag NPs, which are favored by increasing the irradiation time. This fact could be beneficial to reduce Ag oxidation onto the ZnO surface, which contributes to SERS activity. This behavior is observed later.

### 3.3. UV-Vis Absorption

Figure 6 shows UV-Vis absorption spectra of ZnO MSs and Ag/ZnO MSs synthetized with 1.9 (blue lines) and 9.8 mM (red lines) of AgNO_3_ during 5, 15, and 30 min of photodeposition time. At 371 nm, the absorption band associated to the ZnO MSs is observed. It is easy to observe that absorption intensity is increased when the Ag amount is increased as AgNO_3_ concentration or the photodeposition time increase, similar to the behavior reported by other authors [13,14,15]. It reveals that the low incorporation of Ag into ZnO causes a very weak excitonic band with a broad absorption in the visible region (400–550 nm), which is associated to the surface plasmon resonance in the metallic Ag NPs. When the Ag amount is enough high it is possible to see a band clearly [14].

### 3.4. Raman Spectroscopy

Figure 7 shows Raman spectrum of ZnO MSs before Ag deposition, where it is possible to identify peaks at 99, 203, 333, 380, 410, 438, 536, 584, and 657 cm^−1^ labelled as E_2_ (low), 2E_2_ (low), 2E_2_ (low)-2E_2_ (low), A_1_ (TO), E_1_ (TO), E_2_ (high), 2B_1_ (low), A_1_ (LO), and TA+LO modes, respectively, which are vibration modes associated to the wurtzite phase of ZnO [16].

After Ag NPs deposition, we can observe a significative change in all Raman spectra (Figure 8a). Similar behavior has been reported by other authors [17,18,19,20]. The region from 50 to 700 cm^−1^ was deconvoluted to the six identified vibration modes (Figure 8b), of which two belong to Ag deposition. The first one, localized at 220 cm^−1^, corresponds to Ag-O_2_ bonds by chemisorption process due to surface defects in metallic silver [18]. The second one, at 486 cm^−1^, is directly attributed to the interfacial surface phonon mode originated by Ag deposition on the ZnO surface [19]. For this case, Figure 9 shows the Raman intensity from this phonon mode as a function of growth conditions (precursor molar concentration and irradiation time). As we can see, the Raman intensity is directly proportional to precursor concentration and irradiation time, suggesting that the Ag/ZnO sample obtained with 9.8 mM in 30 min, possesses the highest Ag contents.

### 3.5. Detection of R6G with SERS Substrates

SERS spectra of R6G (10^−4^ M) using ZnO MSs and Ag/ZnO MSs at different growth conditions are shown in Figure 10. Pure ZnO and Ag/ZnO samples obtained at 1.9 and 9.8 mM are represented in Figure 10a,b, respectively. On pure ZnO and Ag/ZnO MSs (5 min and 1.9 mM) Raman signal associated to R6G is not identified, due to Ag absence and lower Ag content, respectively. However, for the other samples, clearly defined peaks can be seen at 154, 235, 308 374, 533, 607, 613, 774, 1123, 1182, 1312, 1358, 1508, and 1571 cm^−1^, which are associated to R6G according to other reports [21,22,23,24]. Moreover, it is evident that the SERS activity of the Ag/ZnO substrates improves by increasing the UV irradiation time. To further reveal this behavior, the Raman intensity of the phonon mode at 1506 cm^−1^ versus the irradiation time, using 1.9 and 9.8 mM from the precursor concentration, is represented in Figure 11. It is clear that the Raman signal is benefited with an increment of irradiation time at the highest precursor concentration, so, the sample grown at 9.8 mM during 30 min was chosen to sense the MB molecule at low concentrations due to its highest sensitivity. These results are presented below.

### 3.6. Nanomolar Detection of MB

In order to confirm the detection limit from our substrate with the highest SERS activity, according to R6G sensing, the SERS spectra of MB at different molar concentrations (from 10^−6^ to 10^−9^ M) on Ag/ZnO MSs obtained at 9.8 mM of AgNO_3_ during 30 min of UV irradiation are shown in Figure 12. At 10^−6^ M, we can observe several characteristic peaks from the MB molecule (in spite of the fluorescence band appearing from 300 to 900 cm^−1^, approximately) localized at 243, 441, 482, 610, 1028, 1122, 1436, 1513, 1594, and 1619 cm^−1^, which are assigned to (Ag-N), (C-N-C), (C-N-C), (C-S-C), (C-H), (C-H), (C-N), (C-C), (C-C) ring, and (C-C) ring bands, respectively [25,26]. In addition, other peaks without band assignment are localized at 771, 888, 1069, 1243, 1313, and 1331 cm^−1^ [26]. For lower molar concentrations of MB, the fluorescence was attenuated and some MB peaks disappear when the MB concentration decreases; however, at 10^−9^ M, (Ag-N), (C-N-C), and (C-S-C) bands are clearly identified, making the nanomolar detection of MB possible.

## 4. Conclusions

In this work, we developed satisfactorily hierarchical ZnO MSs and Ag/ZnO MSs, employing hydrothermal and photodeposition methods, respectively (due to their simplicity and low cost). The structural and optical properties of pure ZnO and Ag/ZnO was evaluated according to their growth conditions (precursor concentration and UV irradiation time). XRD results showed for both Ag precursor concentrations (1.9 and 9.8 mM) a strong crystalline quality of Ag dependence over UV irradiation time. All samples were used as SERS substrate to observe their sensitivity to detect R6G at 10^−4^ M. The Raman signals showed that SERS activity of the Ag/ZnO substrates were improved with the increment of precursor concentration and UV irradiation time. For this reason, our Ag/ZnO MSs grown at 9.8 mM in 30 min were selected as a more effective SERS substrate to MB detection at lower concentrations (10^−6^–10^−9^ M). According to TEM images, these powders contain Ag spherical nanoparticles around 35 nm on the ZnO surface that contributed to obtain a nanomolar limit detection (10^−9^ M) of MB.

Our results present a facile method of fabricating effective SERS substrates and the possibility to explore different UV irradiation times using a low molar concentration from the precursor (9.8 mM) to enhance the SERS signal for R6G detection (as the test molecule) and reach low limit detections for other organic colorants, which we could demonstrate with MB.

## Figures and Tables

**Figure 1 nanomaterials-11-03414-f001:**
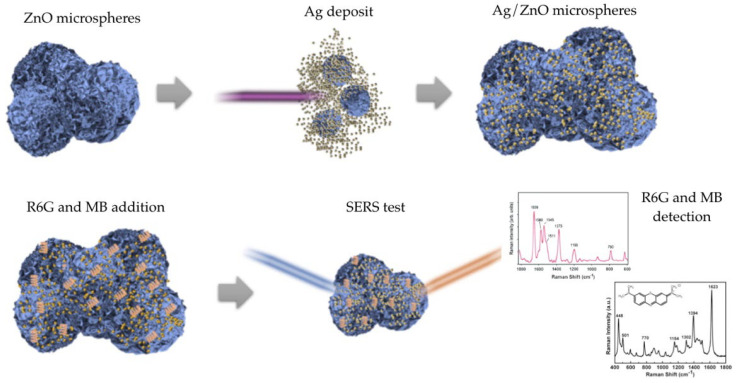
Hierarchical Ag/ZnO MSs based SERS substrates for organic molecules detection.

**Figure 2 nanomaterials-11-03414-f002:**
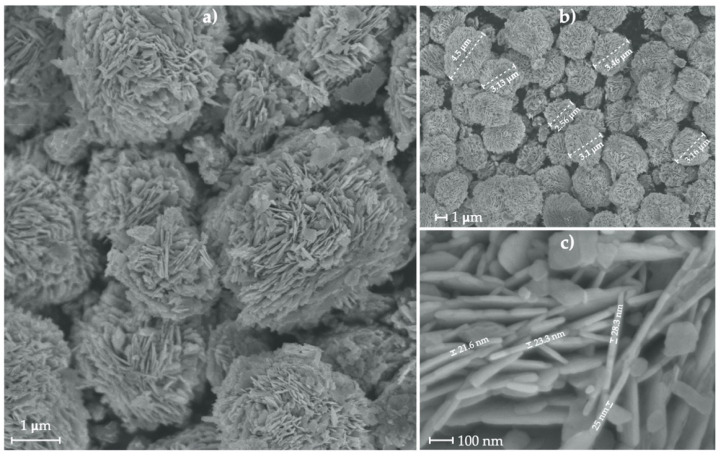
SEM image of ZnO MSs (**a**,**b**) and their two-dimensional nanosheets (**c**).

**Figure 3 nanomaterials-11-03414-f003:**
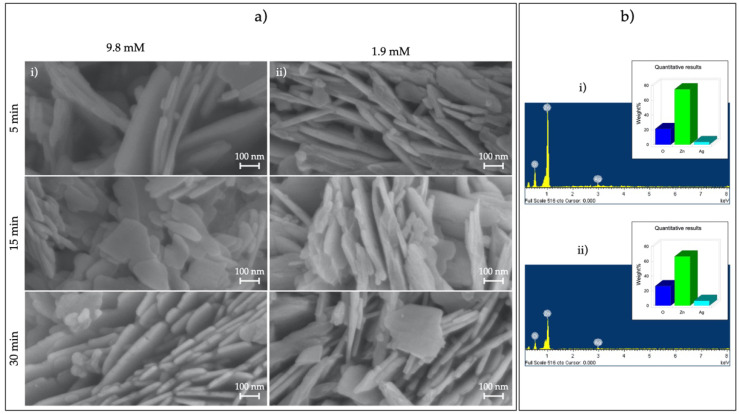
(**a**) SEM images of Ag/ZnO MSs obtained with (**i**) 9.8 and (**ii**) 1.9 mM AgNO_3_ and 5, 15, and 30 min of radiation time and (**b**) EDS of Ag/ZnO MSs obtained at 5 min with (**i**) 9.8- and (**ii**) 1.9-mM silver nitrate.

**Figure 4 nanomaterials-11-03414-f004:**
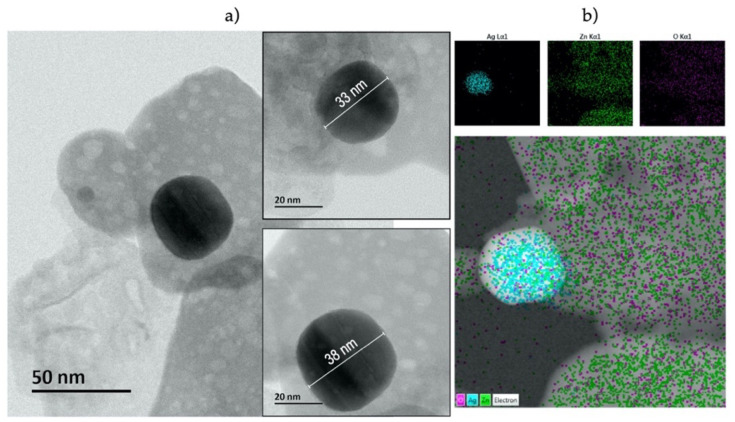
TEM images (**a**) and EDS mapping (**b**) of Ag/ZnO MSs obtained at 9.8 mM of AgNO_3_ during 30 min.

**Figure 5 nanomaterials-11-03414-f005:**
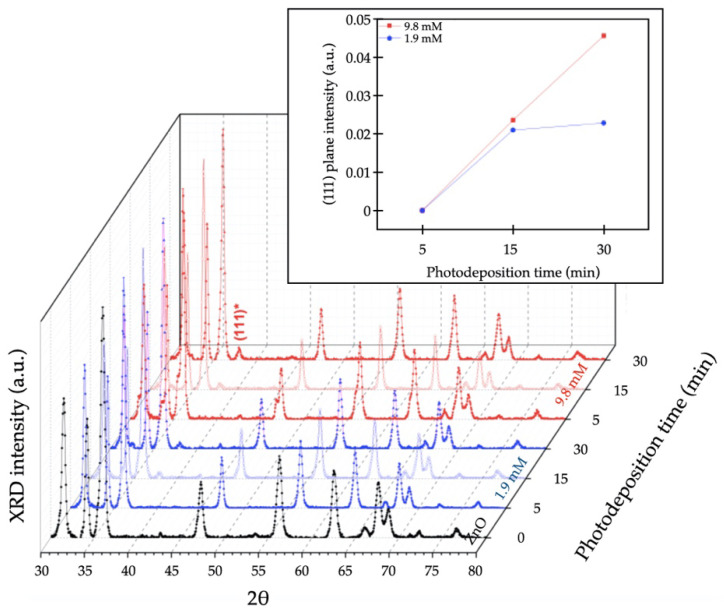
XRD pattern of ZnO MSs (black line) and Ag/ZnO MSs synthetized at 1.9 (blue lines) and 9.8 mM (red lines) of AgNO_3_ during 5, 15, and 30 min. The inset shows the intensity variation of the (111) plane associated to the Ag NPs.

**Figure 6 nanomaterials-11-03414-f006:**
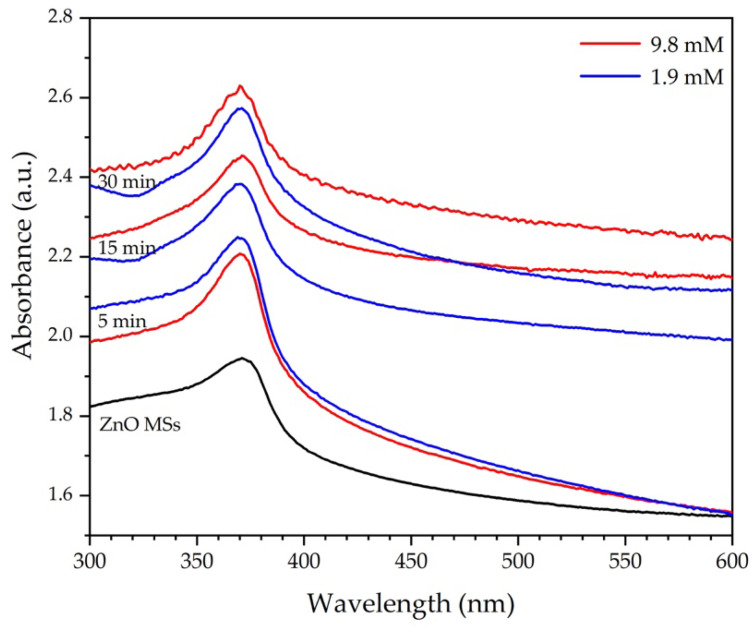
UV-Vis absorption spectra of ZnO MS (black line) and Ag/ZnO MSs obtained at 1.9 (blue lines) and 9.8 mM of AgNO_3_ (red lines), at 5, 15, and 30 min of photodeposition.

**Figure 7 nanomaterials-11-03414-f007:**
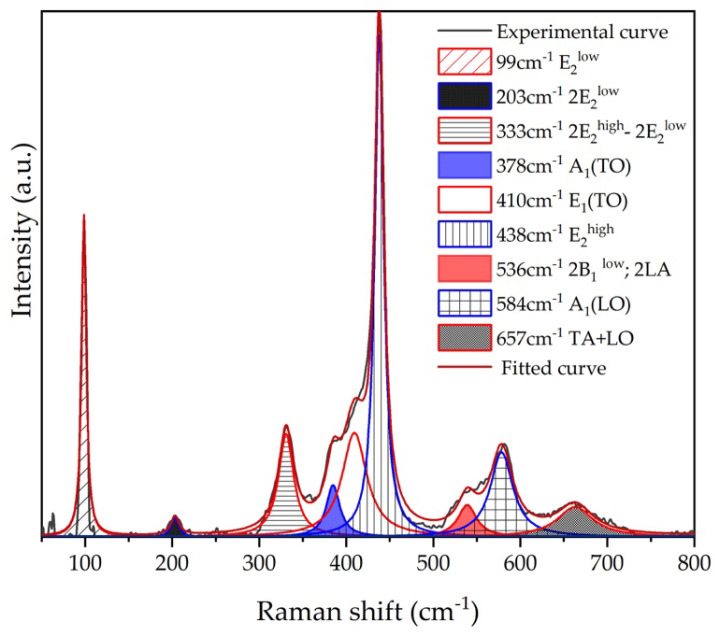
Raman spectrum of ZnO MSs with different localized vibrational modes.

**Figure 8 nanomaterials-11-03414-f008:**
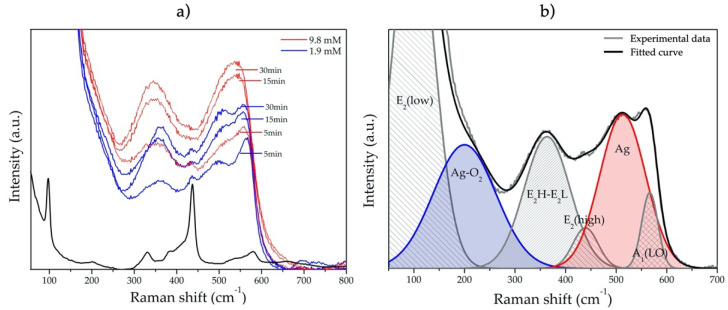
(**a**) Raman spectra of the ZnO MSs (black line) and Ag/ZnO MSs obtained at 1.9 (blue lines) and 9.8 mM of AgNO_3_ (Red lines), at 5, 15, and 30 min of photodeposition. (**b**) Gaussian fit of Raman spectrum of Ag/ZnO MSs obtained at 1.9 mM during 30 min.

**Figure 9 nanomaterials-11-03414-f009:**
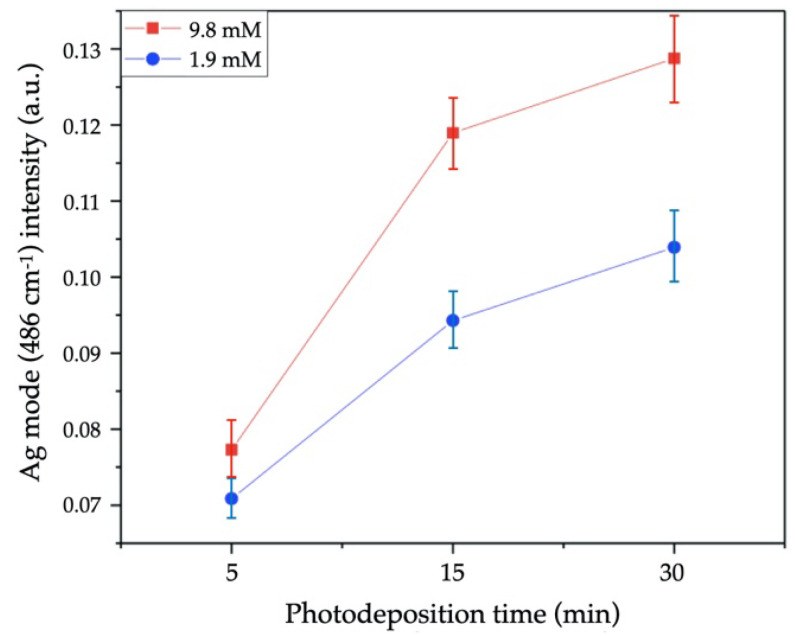
Raman intensity of 486 cm^−1^ mode of Ag/ZnO MSs synthetized at 1.9 (blue line) and 9.8 mM of AgNO_3_ (red line) during 5, 15, and 30 min of UV irradiation time.

**Figure 10 nanomaterials-11-03414-f010:**
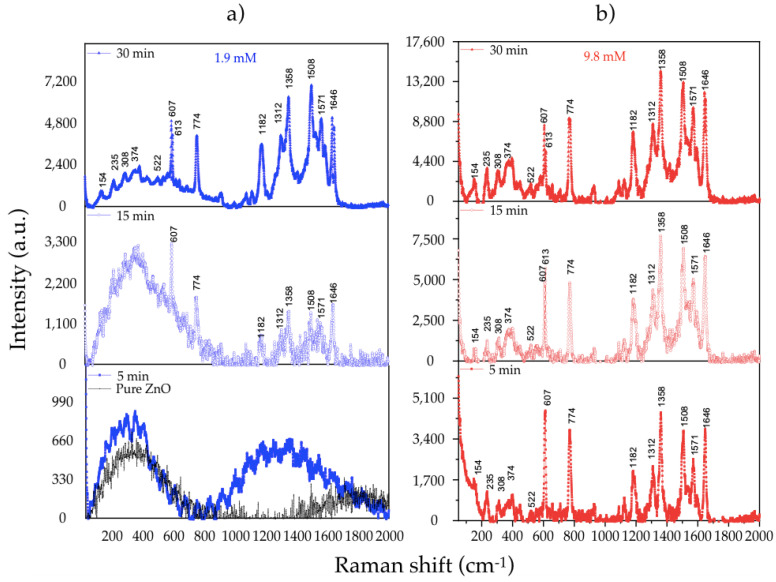
Raman spectra of R6G for (**a**) pure ZnO and using 1.9 and (**b**) 9.8 mM of AgNO_3_ during 5, 15, and 30 min.

**Figure 11 nanomaterials-11-03414-f011:**
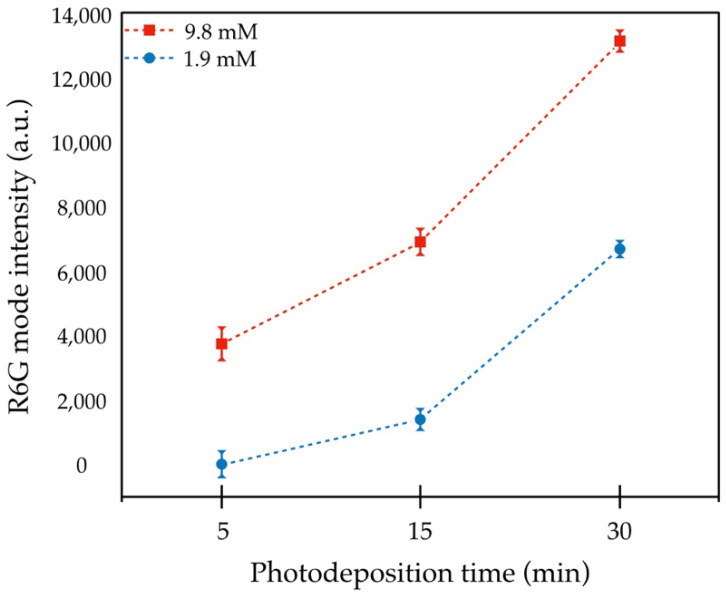
Raman intensity of R6G mode at 1508 cm^−1^ as function of photodeposition time at 1.9 and 9.8 mM of AgNO_3_.

**Figure 12 nanomaterials-11-03414-f012:**
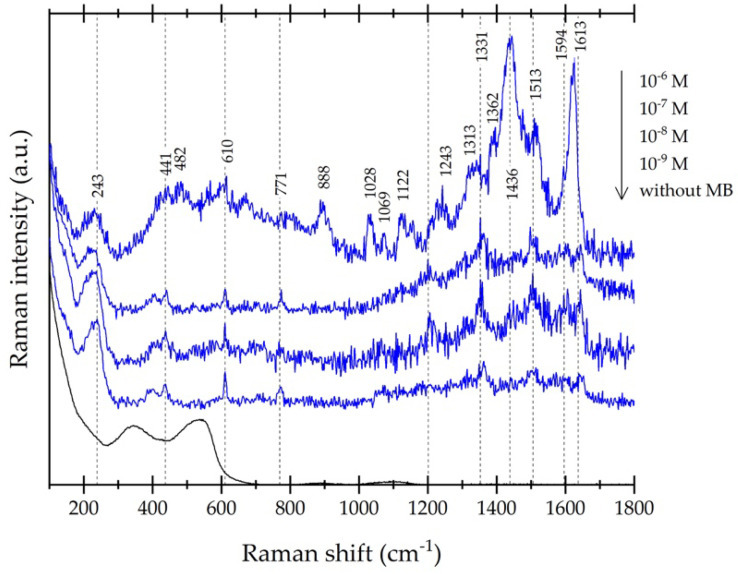
Raman spectra of MB at different concentrations (from 10^−6^ M to 10^−9^ M) measured on Ag/ZnO MSs (9.8 mM of AgNO_3_ during 30 min) and Ag/ZnO MSs substrate without MB.

## Data Availability

The data presented in this study are available on request from the corresponding author.

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
