# Peer review of "Influence of Ag Photodeposition Conditions over SERS Intensity of Ag/ZnO Microspheres for Nanomolar Detection of Methylene Blue"

_nanomaterials, 2021, doi:10.3390/nano11123414_

Round 1
Reviewer 1 Report
The authors have solved all issues/doubts concerning their work.
Author Response
We greatly appreciate all suggestions and comments to improve the paper content
We attended your suggestion about English language; we adjusted the grammar to improve the general presentation.
Reviewer 2 Report
In this work, hierarchical ZnO microspheres (ZnO MSs) and Ag/ZnO MSs were fabricated using hydrothermal and photodeposition methods. In addition, the performance of these potential SERS substrates was evaluated via the detection of R6G and methylene blue. In generally, this work is well organized and written. And it can be considered for publication after some issues are well clarified or revised.
- What is the enhanced factor of the proposed SERS substrate?
- Have authors evaluated the property of recyclable and self-cleaning for the proposed Ag/ZnO MSs-based SERS substrates?
- How is the repeatability of SERS signals using this SERS substrate?
- How is about the storage stability of this SERS substrate?
- The quality of many figures should be improved.
Author Response
We greatly appreciate all suggestions and comments to improve the paper content. The modifications and answers are described below.
- What is the enhanced factor of the proposed SERS substrate?
We don´t have this information. For this case, we opt for report the limit detection from our SERS substrate (for methylene blue) to compare with other nanostructures.
- Have authors evaluated the property of recyclable and self-cleaning for the proposed Ag/ZnO MSs-based SERS substrates?
Not yet. Until now, we were evaluated only the sensitivity from our SERS substrates. We are considering this important property to next works.
- How is the repeatability of SERS signals using this SERS substrate?
We added the sentence "Three samples for each growth conditions were measured in 10 different points" in lines 144 and 145. And we added the error bars in Fig. 10.
- How is about the storage stability of this SERS substrate?
We didn’t evaluate the stability from our substrates until now. We are considering the stability for next work.
- The quality of many figures should be improved.
Thanks for you suggest. Fig 1, 6, 7 and 10 were improved.
Thank you for your valuable time,
Kind regards,
Reviewer 3 Report
The authors describe a SERS substrate synthesized from ZnO microspheres and Ag deposition via UV irradiation of Ag+. While Ag/ZnO MSs are not novel, the authors describe the advantages of their approach as being easier, cheaper and greener than previous methods. The authors investigated the concentration of Ag+ precursor and the length of time of UV irradiation and find, unsurprisingly, that the more concentrated precursor and longer exposure times lead to more Ag deposition and a higher SERS signal.
The paper requires only some moderate editing for English and may be of interest to the journal's readership. Answers to the following questions might provide some additional depth:
1) The authors use EDS to measure the Ag content of the substate. Fig. 3 includes only the 5 minute UV exposure for the two precursor concentrations. Later in the paper, they use the Ag Raman mode (483 cm-1) to show that the higher precursor concentration and longer UV irradiation times lead to a larger Ag content. If the authors are using EDS for a similar characterization, why not show the EDS data for the longer irradiation times to corroborate the Raman data?
2) The authors indicate in the introduction that the crystallinity of the Ag is important for mitigating the problem of oxidation of Ag, an important, well known limitation of Ag SERS substates. The authors show XRD data that indicate increasing (111) intensity for increasing precursor concentration and UV time. It it not clear to me if the increased (111) intensity is due to more crystallinity or simply more Ag being deposited (See comment 1). Some additional discussion on the crystallinity would be beneficial.
3) Related to Question 2), the authors do not discuss the stability of the substate (even though they allude to the crystallinity being an important factor). How does the SERS response change over time due to oxidation?
4) How does the SERS response compare for substrates in this work compare with similar Ag substrates with and without the ZnO MS base? That is, the authors indicate which of their substates performed better, but how does their best structure compare with other similar substrates in the literature?
Author Response
- The authors use EDS to measure the Ag content of the substrate. Fig. 3 includes only the 5minute UV exposure for the two precursor concentrations. Later in the paper, they use the Ag Raman mode (483 cm-1) to show that the higher precursor concentration and longer UV irradiation times lead to a larger Ag content. If the authors are using EDS for a similar characterization, why not show the EDS data for the longer irradiation times to corroborate the Raman data?
We understand your doubt. Initially, we didn’t have a characterization technique that confirmed the Ag presence on ZnO surface. Subsequently, we measured TEM and EDS mapping to our sample that showed best SERS activity (according to Fig 9 and 10). After that, we chose this sample to complete the methylene blue detection and we had the opportunity to obtained SEM-EDS measurements for 2 samples. We selected the samples with lower precursor concentration and irradiation time, inferring that if was possible to observe Ag content in this samples, the other samples should have more Ag content. Finally, the study about Ag content was completed with Raman data. We hope you understand the restrictions to access other measurements (pandemic).
- The authors indicate in the introduction that the crystallinity of the Ag is important for mitigating the problem of oxidation of Ag, an important, well known limitation of Ag SERS substates. The authors show XRD data that indicate increasing (111) intensity for increasing precursor concentration and UV time. It it not clear to me if the increased (111) intensity is due to more crystallinity or simply more Ag being deposited (See comment 1). Some additional discussion on the crystallinity would be beneficial.
We appreciate your suggestion. The increasing (111) plane intensity is due to more crystallinity. We added more information in the X-ray diffraction section (178-184 lines).
- Related to Question 2), the authors do not discuss the stability of the substate (even though they allude to the crystallinity being an important factor). How does the SERS response change over time due to oxidation?
We didn’t evaluate the stability from our substrates until now. We are considering the stability for next work.
- How does the SERS response compare for substrates in this work compare with similar Ag substrates with and without the ZnO MS base? That is, the authors indicate which of their substates performed better, but how does their best structure compare with other similar substrates in the literature?
In order to confirm the SERS response from our best substrate (MB limit detection: 10-9 M) with other similar works. We share to you this table. All works are based on chemical routes to obtain their SERS substrates to detect MB (like us). Several works with Ag/ZnO are based in physical methods and other ones are focused on degradation processes, but they are not comparable with our contribution.
|
Year of publication |
SERS substrate (materials) |
ZnO morphology |
MB limit detection |
|
2021 [1] |
ZnO |
Plates, hexagonal, tube, flory-rod |
10-4 M |
|
2018 [2] |
Au/ZnO |
Nanorods |
10-6 M |
|
2018 [3] |
Au/ZnO |
Nanorods |
10-7 M |
|
2021 [4] |
Au/ZnO |
Nanorods |
10-9 M |
|
2019 [5] |
Au/ZnO |
Nanorods |
10-9 M |
[1] Optical Materials 120 (2021) 111460. https://doi.org/10.1016/j.optmat.2021.111460
[2] Surface Review and Letters, 25, 1840004 https://doi.org/10.1142/S0218625X18400048
[3] AIP Conference Proceedings, 2010, 020023 https://doi.org/10.1063/1.5053199
[4] Applied Surface Science, 554, 149653 https://doi.org/10.1016/j.apsusc.2021.149653
[5] Talanta 194 (2019) 680–688 https://doi.org/10.1016/j.talanta.2018.10.060
As you can see, Au/ZnO substrates implies high costs of fabrication respect our fabrication method. Furthermore, we get an optimization process as we report in introduction section.
Thank you for your valuable time,
Kind regards,
Reviewer 4 Report
Comments: The manuscript reports on a preparation and characterization of ZnO microspheres coated with silver nanoparticles as potential SERS active substrates. Preparation procedure under different experimental conditions was demonstrated in detail and the prepared ZnO and Ag/ZnO microspheres characterized using microscopy (SEM, TEM) and X-ray diffraction method. Finally, the SERS activity of the Ag/ZnO microspheres was tested using rhodmine 6G (R6G) and methylene blue (MB). Though preparation and characterization of the Ag/ZnO microspheres are clearly described, there are some issues regarding surface-enhanced Raman scattering which should be addressed before publication.
Remarks:
- In my opinion the title of the manuscript is too long and not quite clear. It should be more concise.
- For newly prepared metallic nanostructures (Ag/ZnO MSs), potentially used as the SERS active substrates, it is necessary to measure plasmon frequency. This characteristic is crucial for the surface enhancement of the Raman scattering and very important with regard to the choice of the excitation wavelength (laser) and interpretation of the obtained spectra.
- Why was MB used to test sensitivity of the Ag/ZnO MSs, and not R6G?
- The SERS spectrum of MB at concentration of 10-6 M and the spectra of lower MB concentrations (10-7 M, 10-8 M) differed, not only in intensity but in the observed bands. What was the cause of the obtained spectral differences? Could they be the result of the concentration induced changes in position and/or orientation of MB molecules with respect to the silver surface? Please explain.
- Given very weak intensity of the bands in the spectrum of MB at 10-9 M, it would be useful to compare the spectrum of Ag/ZnO microspheres without MB (blank) with the spectra of low concentration MB samples in Fig 11., in order to confirm that very weak bands originate from MB.
- Was the reproducibility of the substrates and SERS measurements checked? If yes, in what way?
- Authors are advised to check and correct the English language.
Author Response
We greatly appreciate all suggestions and comments to improve the paper content. The modifications and answers are described below.
- In my opinion the title of the manuscript is too long and not quite clear. It should be more concise.
We attended your suggestion. We proposed to editor the title: “Influence of Ag photodeposition conditions over SERS intensity of Ag/ZnO microspheres for nanomolar detection of methylene blue”
- For newly prepared metallic nanostructures (Ag/ZnO MSs), potentially used as the SERS active substrates, it is necessary to measure plasmon frequency. This characteristic is crucial for the surface enhancement of the Raman scattering and very important with regard to the choice of the excitation wavelength (laser) and interpretation of the obtained spectra.
The plasmon frequency was not measured directly; however, according to TEM images, the Ag NPs sizes are around 30 nm, so, the plasmon frequency can be around 420 nm. For this reason, we used a 532 nm laser in Raman spectroscopy.
- Why was MB used to test sensitivity of the Ag/ZnO MSs, and not R6G?
In previously review of this work, we received a suggestion to add another compound (in addition to R6G) to measure the SERS activity from our substrates. We decided to evaluate this activity with R6G and complete the sensitivity with methylene blue (limit detection).
- The SERS spectrum of MB at concentration of 10-6 M and the spectra of lower MB concentrations (10-7 M, 10-8 M) differed, not only in intensity but in the observed bands. What was the cause of the obtained spectral differences? Could they be the result of the concentration induced changes in position and/or orientation of MB molecules with respect to the silver surface? Please explain.
In MB at 10-6 spectrum, there are superposition peaks originated by fluorescence, when fluorescence disappears or is attenuated, the peaks than composed some “bands” are clearly identified at lower MB concentrations. When MB concentrations are decreasing, some peaks are suppressed; but all peaks are originated exclusively by MB.
- Given very weak intensity of the bands in the spectrum of MB at 10-9 M, it would be useful to compare the spectrum of Ag/ZnO microspheres without MB (blank) with the spectra of low concentration MB samples in Fig 11., in order to confirm that very weak bands originate from MB.
Thanks for you suggest. We show below the Ag/ZnO and MB (10-9 M) on Ag/ZnO substrate Raman peaks positions to confirm that peaks on Fig. 11 correspond to MB.
Ag/ZnO: 99, 220, 333, 438, 486 and 584 cm-1
MB (10-9 M) on Ag/ZnO: 243, 441,610, 771, 1331, 1513, and 1619 cm-1
- Was the reproducibility of the substrates and SERS measurements checked? If yes, in what way?
We added the sentence "Three samples for each growth conditions were measured in 10 different points" in lines 144 and 145. And we added the error bars in Fig. 10.
- Authors are advised to check and correct the English language.
We attended your suggestion; we adjusted the grammar to improve the general presentation.
Thank you for your valuable time,
Kind regards.
Round 2
Reviewer 2 Report
Most concerns raised in previous report have not been well addressed.
Author Response
We greatly appreciate all suggestions and comments to improve the paper content. The modifications and answers are described in the PDF file.
We hope that you remarks are been resolved and we appreciate your contribution to improve the manuscript content.
Thank you for you valuable time,
Kind regards,

Reviewer 4 Report
Though authors have answered in written on all my remarks, they have not improved the article in suggested ways.
For example, additional experimental work has not been done in order to measure plasmon frequency of the prepared SERS substrates (remark 2). It is well known that Ag nanoparticles dispersed in aqueous medium are characterized by plasmon frequency in the range 400-420 nm, but when they aggregate or form a coating/layer on a solid support, the plasmon frequency shifts to higher wavenumbers, depending on the layer structure and chemical composition of the solid support. Thus, very important data on studied substrates are missing.
Further on, additional, more detailed explanation about differences in concentration dependent MB spectra could have been added in the text (remark 4). Moreover, the broad baseline (10-6 M) could be the result of fluorescence, but not the observed vibrational bands.
Finally, the authors could insert the spectrum of Ag/ZnO microspheres without MB in Fig. 11 in comparison to spectrum of MB (10-9 M) to prove spectral differences.
I am sorry, but I am not convinced by the presented study and I cannot recommend the publication of the revised version.
Author Response

(The authors gave the same response as above.)

Round 3
Reviewer 2 Report
The current version can be accepted.